# Relationship between Physical Activity, Sedentary Behavior, and Anthropometric Measurements among Saudi Female Adolescents: A Cross-Sectional Study

**DOI:** 10.3390/ijerph18168461

**Published:** 2021-08-10

**Authors:** Abeer Ahmad Bahathig, Hazizi Abu Saad, Nor Baizura Md Yusop, Nurul Husna Mohd Shukri, Maha M. Essam El-Din

**Affiliations:** 1Department of Nutrition, Faculty of Medicine and Health Science, Universiti Putra Malaysia, UPM Serdang, Selangor 43400, Malaysia; abeer-ahmad-b@hotmail.com (A.A.B.); n_husna@upm.edu.my (N.H.M.S.); 2Department of Nutrition and Food Science, College of Home Economic, Northern Border University, Arar 91431, Saudi Arabia; Mahaaessam82@gmail.com; 3Department of Dietetics, Faculty of Medicine and Health Science, Universiti Putra Malaysia, UPM Serdang, Selangor 43400, Malaysia; norbaizura@upm.edu.my; 4Department of Nutrition and Food Science, Faculty of Home Economics, Helwan University, Cairo 11790, Egypt

**Keywords:** obesity, overweight, sociodemographic characteristics, waist to height ratio, health behavior

## Abstract

Overweight and obesity are becoming increasingly prevalent among children and adolescents in Saudi Arabia and are an emerging cause of non-communicable diseases (NCDs). Lifestyle factors, such as insufficient levels of physical activity and sedentary behaviors, are responsible for the increased prevalence of NCDs. This study aimed to determine the association between physical activity levels, sedentary behaviors, and anthropometric measurements in Saudi female adolescents. A cross-sectional study was carried out among 399 healthy female adolescent students aged 13–14 years in Arar, Saudi Arabia. The participants were randomly selected from different schools and their anthropometric measurements were determined. The Physical Activity Questionnaire for Older Children (PAQ-C) and the Adolescent Sedentary Activity Questionnaire (ASAQ) were used to assess their physical activity levels and sedentary behaviors, and an analysis was conducted using IBM SPSS software version 25. A multiple linear regression model was used to determine the association between the variables. The majority of the participants had a normal body mass index (BMI; 79.4%) and waist circumference (WC; 62.4%). A total of 74.4% had waist to height ratio (WHtR) < 0.5. About 92.7% of the participants were not meeting PA recommendations of 60 min of moderate to vigorous physical activity daily. The overall mean time spent on sedentary activities was high on both weekdays and weekend days at 357.64 ± 86.29 and 470.51 ± 147.64 min/day, respectively. Moreover, anthropometric measurement (BMI) was positively associated with age and negatively associated with sedentary behavior on weekends, while WHtR was positively associated with age. The multiple linear regression analysis also showed that age and sedentary behavior significantly predicted BMI among the study participants (F (2, 396) = 4.346, *p* < 0.014) and age was the only significant predictor of WHtR (F (1, 397) = 16.191, *p* ≤ 0.001). This study revealed that most of the female Saudi adolescents undertook low levels of activity and high levels of sedentary behaviors. Sedentary behaviors were significantly associated with their BMI. Accordingly, an intervention program on healthy lifestyles is important to improve Saudi female adolescents’ lifestyles.

## 1. Introduction

Overweight and obesity place people at a high risk of non-communicable diseases (NCDs) [1] and health problems worldwide [2]. Overweight and obesity among schoolchildren and adolescents are a major public health challenge facing societies in many parts of the world [3,4]. Children and adolescents suffer from overweight and obesity in both developed and developing countries [5], and the international prevalence has risen within the period 1975–2016 from 4% to >18% in adolescents of both genders [6]. Globally, 340 million individuals aged 5–19 years were overweight in 2016 [6] and nearly 13 million aged 2–19 years were obese [7]. The overall prevalence of obesity in Saudi Arabia was estimated to be 35.4% compared with 31.7% in the United Arab Emirates, 27.8% in Syria, 27% in Oman and 30.4% in Iraq [8]. In Saudi Arabia, the prevalence of overweight and obesity has been found to be higher among girls than boys [9,10] and is increasing remarkably faster than in many other countries in the region [11,12,13,14]. Findings in 2010 relating to Saudi boys and girls demonstrated that more girls tended to be overweight compared to boys (28.4% vs. 24.8%, respectively), and the prevalence of obesity was also found to be higher among girls than boys (11.2% vs. 10.0%, respectively) [15]. Overweight and obesity in children have been associated with the risk of cardiovascular diseases, such as hypertension, as well as metabolic syndrome and diabetes mellitus [16,17,18].

The prevalence of insufficient levels of physical activity has increased in Saudi societies in recent decades due to economic development and technological advancements, which explain the reason most Saudis, including children, youths, and adults, frequently do not meet the required daily level of physical activity, even during their leisure time [19]. The prevalence of insufficient levels of physical activity was found to be >40% in Arab countries, for example, 87% in Sudan and 68% in Saudi Arabia [20]. Persistent low levels of physical activity are known to be detrimental to individual health and well-being [21,22,23] and are linked to less healthy lifestyles [24]. Information about the patterns of physical activity among Saudi adolescents is currently insufficient, so further research is needed to gain more insights into Saudi physical activity levels as they relate to overweight and obesity.

Importantly, high levels of sedentary behaviors and insufficient levels of physical activity increase obesity among both children and adolescents [25], and raise the risk of morbidity and mortality, cardiovascular diseases [26], and type 2 diabetes [27] among adolescents. Sedentary behaviors and insufficient levels of physical activity were separately and independently associated with metabolic risk and obesity [28]. Sedentary activity has also been linked with dietary habits and unhealthy lifestyle choices [29,30]. Tremendous lifestyle changes have taken place in Saudi Arabia over the past few decades, and sedentary behaviors are becoming more prevalent among Saudi children and adolescents [31]. Alzami et al. [21] showed that more than 85% of Saudi female adolescents spend >3 h per day sedentary. Physical activity, sedentary behavior, and body weight status among adolescents are important issues because these factors are associated with their health—“a state of complete physical, mental and social well-being and not merely the absence of disease or infirmity” [32]. A previous study showed that physical activity could affect the quality of life of students aged 12 to 15 years old through self-concept and subjective happiness [33]. Based on the findings of a multivariate analysis, physical activity was also proven to be one of the significant factors positively associated with overall health-related quality of life [34]. Moreover, the impact of body weight on quality of life was also found to be significant, based on findings that indicated that the normal-weight group had significantly higher scores of health-related quality of life compared to the obese group [35]. This study, therefore, aimed to determine the relationship between physical activity levels, sedentary behaviors, and anthropometric measurements among Saudi female adolescents aged 13–14 years. We hypothesized that the prevalence of insufficient physical activity and sedentary behaviors are high among Saudi female adolescents in this age group. We also hypothesized that physical activity and sedentary behaviors are significantly associated with adolescent anthropometric measurements.

## 2. Materials and Methods

### 2.1. Study Design and Location

A cross-sectional study was conducted in Arar, which is on the northern border of Saudi Arabia, between September and October 2019. The study participants were randomly selected using a multistage random sampling technique at four government schools. All 13 areas in Saudi Arabia were included before the northern border region was selected. This was followed by the selection of the city of Arar from four cities in this region. Arar was then further divided into five areas, and four regions were subsequently selected, namely, the northern, central, eastern, and western regions. One school was selected from each of these regions. Accordingly, four female public intermediate schools were randomly selected out of 22 schools in Arar.

### 2.2. Ethical Clearance

Permission to conduct this study was granted by the Local Committee of Bio Ethics (HAP-09-A-043) at Northern Border University Ref No (13/40/H) dated: 13 March 2019), and the Ministry of Education in Arar, Saudi Arabia. Moreover, the University Ethics Committee for Research involving Human Subjects of the Universiti Putra Malaysia (UPM) approved the study to be carried out with reference number: UPM/TNCPI/RMC/JKEUPM/1.4.18.2(JKEUPM) dated: 23 October 2019. Written informed consent was obtained from all the participants and their parents. The participants’ information was kept confidential. The entire development of the study followed the ethical recommendations determined by the Declaration of Helsinki.

### 2.3. Study Participants

The female students aged 13–14 years who agreed to participate in the study and whose parents provided their consent were recruited for the study. Students with medical conditions, such as asthma, diabetes, cancer, cardiovascular diseases, fractures, cirrhosis, or other diseases for which they were receiving treatment, and those who were physically disabled, were excluded from the study.

### 2.4. Sample Size Determination

The formula for calculating sample size for one group cross-sectional study was employed in this study, as shown below [36]:$$n=\frac{z^{2}1-\frac{\alpha }{2}p\left(1-p\right)}{d^{2}}$$
where: 

*n* = estimated number of samples.

*z*^2^1 − α/2 = standard error when α = 0.05 (95% Confidence Interval) = 1.96.

*p* = prevalence of overweight and obesity among young female adolescents aged between 12 and 14 years in Saudi Arabia = 23.3% = 0.233 [37].

*d* = desired precision was at 5% = 0.05.

By substituting the values in the above formula and adding a 30% attrition rate, the minimum sample size needed for this study would be 366. After approaching the selected schools, a total of 493 participants were invited, and 399 agreed to participate. Therefore, the response rate for this study was 80.9%. Among the reasons given for not being included in the study were: not being available at the time of data collection, and not being willing to participate in this study.

### 2.5. Data Collection

The students were given a questionnaire, which they were asked to complete with face-to-face guidance from the researcher. The questionnaire comprised the following sections: sociodemographic characteristics, physical activity, and sedentary behaviors. The anthropometric indicators of all the students were obtained by the researcher.

#### 2.5.1. Sociodemographic Data

The sociodemographic data included age, date of birth, number of siblings and household members, school location, mother’s and father’s education levels, and monthly income.

#### 2.5.2. Anthropometric Measurements

A digital body weighing scale (Detecto SOLO Digital Clinical Scales, Webb City, MO, USA) was used to measure the participant’s body weight to the nearest 0.1 kg and height to the nearest 0.1 cm. The participants were required to wear light clothes, to keep their head upright, arms at their sides and shoulders relaxed and to stand up straight with their back against the stadiometer rule. Body mass index (BMI)-for-age (z-score; kg/m^2^) was measured based on a growth reference for girls aged 5–19 years with the following categories: thinness (<−2 standard deviation [SD]), normal weight (≤−2 SD and ≥1 SD), overweight (>+1 SD) and obesity (>+2 SD) [38]. WC was measured using a tape to the nearest 0.1 cm at the point between the top of the iliac crest and the lower part of the last rib (WHO, 2008), with the cut-off points of 72.3 cm (overweight) and 77 cm (obesity) [39]. WHtR was calculated as WC (cm)/height (cm), and abdominal obesity among adolescents was identified using a suggested WHtR cut-off of ≥0.50 [40]. All the measurements were performed twice for accuracy, and the mean value was used for further analysis.

#### 2.5.3. Physical Activity

The Physical Activity Questionnaire for Older Children (PAQ-C), which comprises nine items, was used in this study. PAQ-C is a 7-days physical activity recall questionnaire assessing activities during weekdays and at the weekend. The questionnaire assessed the subject’s physical activity during spare time, physical education classes, activity at recess, at lunch, right after school, and evening activities. This instrument has been established and validated for school students [41,42] and has a Cronbach’s alpha value of 0.777 [43]. The score of each item is measured on a five-point Likert scale from low physical activity (1) to high physical activity (5). The final PAQ-C score is the mean score of all the items [41]. Saint-Maurice et al.’s equation was used to calculate the percentage of minutes per day of moderate to vigorous physical activity (MVPA) [44]. Based on the recommendations by the World Health Organization (2020), children and adolescents aged 5–17 years should do at least an average of 60 min per day of MVPA; therefore, participants with ≥420 min of MVPA per week were classified as meeting physical activity recommendations and <420 min of MVPA weekly as not meeting physical activity recommendations [45].

#### 2.5.4. Sedentary Behavior

The Adolescent Sedentary Activity Questionnaire (ASAQ) developed by Hardy et al. [46] was used to determine the sedentary behavior levels of the participants on weekdays and weekend days. The test–retest correlations have previously been established as ≥0.70 [46]. The ASAQ queries the time spent per day carrying out 11 different sedentary behaviors. The categories of the activities are screen time, education, travel and cultural activities, and social activities. The total time of all the categories is collected to determine the overall sedentary time per weekday and per weekend day. In the ASAQ, sedentary behaviors ≥4 h per day are considered high levels of sedentary behaviors, whereas a score of <4 h per day means low levels of sedentary behaviors [47].

### 2.6. Statistical Analysis

The Statistical Package for the Social Sciences for Windows version 25 (Chicago, IL, USA) was used to analyze the data at 95% confidence intervals (CIs). The mean, SD, percent, and frequency were used to express the results. A multiple linear regression model was used to determine the association between the variables. A *p*-value < 0.05 was considered statistically significant.

## 3. Results

### 3.1. Sample of the Study

Table 1 presents the descriptive analysis of the participants’ sociodemographic characteristics. The mean age of the participants was 13.34 ± 0.49 years. Among the participants, 53.1% lived with 5–10 siblings, while 42.4% reported having <5 siblings, with only 4.5% participants indicating that they lived with >10 siblings in their family. The majority (77.2%) of the participants reported having 5–10 family members in their household, and only 4.3% lived with <5 family members in their household. Overall, 54.6% of the fathers and 48.1% of the mothers had studied beyond high school, indicating that the majority of the participants had educated parents, and the percentage of the fathers’ and mothers’ education at each level was almost equal. Half of the participants (49.6%) reported having a total family monthly income of between Saudi Riyals (SAR) 5000 and SAR 14,999, which is considered a middle-income family income.

### 3.2. Participants’ Anthropometric Measurements

Table 2 presents the general anthropometric measurements of the participants, namely, their height, weight, and BMI. The mean of the participants’ height (m) and weight (kg) were 151.3 ± 9.5 cm and 51.1 ± 13.6 kg, respectively. The mean of the BMIs for the age z-score was 22.92 ± 14.16, with a range between 12.55 kg/m^2^ and 22.8 kg/m^2^. The overall mean of the BMI z-scores for all the participants was 0.57 ± 1.39. Most (79.4%) of the participants had a normal weight. The prevalence of obesity and overweight among the participants were 2.5% and 13.0%, respectively. The mean WC was 70.7 ± 10.1 cm, demonstrating that the participants had an average normal WC. While 62.4% of the participants had a normal WC, only 13.0% and 24.6% of the participants were overweight and had an abdominal obesity status based on WC, respectively. According to WHtR, the mean score was 0.46 ± 0.06, and most participants (74.4%) had a normal WHtR.

### 3.3. Level of Physical Activity among the Participants

Table 3 shows the distribution of the levels of the participants’ physical activity. Only 7.3% of the participants were classified as meeting physical activity recommendations, while 92.7% not meeting physical activity recommendations of at least 420 min of MVPA weekly, which means that there was a lack of participation and involvement in physical activity in general among the study population.

### 3.4. Distribution of Respondent’s Sedentary Behaviour by Weeks and Weekends

The overall mean time spent on sedentary behaviors during week and weekend days were 357.64 ± 86.29 min/day and 470.51 ± 147.64 min/day, respectively (Table 4). This means that the participants spent on average six hours on sedentary activities during weekdays and eight hours during weekend days, which is very high. During weekdays, the mean of sedentary screen time was 141.03 ± 58.69 min/day, and the means for education and social activities were 117.89 ± 36.22 min/day and 44.94 ± 18.48 min/day, respectively. However, the time spent engaged in sedentary cultural activities and travel was low (37.02 ± 28.73 and 17.95 ± 8.65 min/day, respectively) during weekdays. In contrast, on weekend days, the mean sedentary screen time was 201.37 ± 97.37 min/day, the mean sedentary education time was 122.78 ± 61.49 min/day, and the mean for sedentary social activities was 68.77 ± 39.89 min/day. However, low levels of sedentary cultural activities and travel were noted, with means of 59.9 ± 46.5 min/day and 23.7 ± 14.8 min/day, respectively. Overall, 90% of the participants participated in sedentary behaviors during weekdays and 95% participated in sedentary behaviors during weekend days.

### 3.5. Factors Associated with Anthropometric Measurements

Table 5 shows the associations between sociodemographic characteristics, physical activity, and sedentary behaviors, with BMI and WHtR calculated using multiple regressions models. The 10 variables initially used in the simple linear regression model were: age (years), number of siblings, number of household members, mother’s education, father’s education, family’s monthly income, total mean score of physical activity, sedentary behavior, total sedentary behavior weekdays, and total sedentary behavior weekends. Only age and sedentary behaviors at weekends had *p*-values less than 0.24, and these were used in the backward stepwise regression analysis. Moreover, these two variables were the only variables that were retained in the final model, and they were found to significantly predict (*p* < 0.05) the anthropometric measurements of the study participants (F (2, 396) = 4.346, *p* < 0.014). A significantly moderate relationship (R = 0.017) was observed between these two predictors. For WHtR, a significant moderate relationship (R = 0.037) was observed with the age variable only: WHtR (F (1, 397) = 16.191, *p* = 0.00.

The final model accounted for 1.7% of the variation in the participants’ anthropometric measurements (BMI). The equation used in the backward stepwise multiple regression model was anthropometric measurements (BMI) = −19.870 + (3.523 age) + (−0.009 sedentary behavior at weekends). This equation showed that the anthropometric measurements were positively associated with age and negatively associated with sedentary behaviors at weekends. While WHtR was positively associated with age (WHtR) = 0.431 + (0.198 age). The multiple linear regression models also indicated that physical activity did not significantly predict the anthropometric measurements (*p* > 0.05).

## 4. Discussion

The present study reported the prevalence of overweight and obesity and their associated factors as it relates to anthropometric measurements among female adolescents aged 13–14 years from Arar in Saudi Arabia. The findings of this study provided evidence of the high prevalence of sedentary behaviors and insufficient levels of physical activity among the study participants. Moreover, the regression analyses revealed that age and unhealthy behaviors were key factors leading to increases in body weight and overweight in the study participants. The majority of the adolescents (92.7%) were not meeting physical activity recommendations of at least 420 min of MVPA weekly, and only 7.3% of the participants were classified as meeting physical activity recommendations. Our results correspond with those of Miranda et al. [48] who showed that 84.2% of Brazilian female adolescents aged 14–19 years had low physical activity levels. Among a cohort of female adolescents aged 13–14 years in Kuantan Pahang, Malaysia, 82.7% were classified as having low physical activity levels [49]. Moreover, a study by Kerkadi et al. [50] conducted among 1184 Qatari adolescents aged 14–18 years reported that 54.2% of the female participants were inactive compared to 33.1% of their male counterparts. The mean BMI and WC based on the MET (minutes/week) from moderate activity were 1115.6 ± 1034.0 and 1196.8 ± 1072.3, respectively. Furthermore, there was a clear association between a low frequency of fruit consumption and low levels of physical activity among 3859 Dutch adolescents (12–18 years of age) [51]. Among them, 20.8% of the participants did not use a bicycle for transport, 42.4% used a bicycle occasionally, and 36.8% always used a bike for transport. The mean activity period for these adolescents was 35.11 (SD = 31.51) min/day.

In contrast to our findings, Alharbi conducted a study in Riyadh among adolescents aged 10–15 years using the PAQ-C and found that 73.5% of the participants reported moderate levels of physical activity, 22.4% low levels of physical activity and 4.1% high levels of physical activity [52]. Similarly, a study of 275 non-Saudi girls aged 9–16 years who lived in Saudi Arabia used the PAQ-C and the Godin–Shephard Leisure-Time Physical Activity Questionnaire (GSLTPAQ) and found that 65.8% were physically active and 34.2% were insouciantly physically active based on the latter questionnaire, while 50% were moderately active and 22.2% were extremely active based on the PAQ-C. Moreover, there were no statistically significant differences in their levels of physical activity, screen time, and BMI status [53]. In comparison, a study found that expatriate non-Saudi children who lived in Saudi Arabia were more physically active than Saudi children of the same gender and age (8–18 years) [54]. In Malaysia, 57.3% of adolescents (male and female) aged 10–17 years were identified as undertaking insufficient levels of physical activity [55]. According to a WHO report that compared gender in terms of physical activity among an adolescent population aged 11–17 years, 80% of the participants were physically inactive, and the female population were less physically active than the male population [56]. The data regarding the activity patterns of younger girls were less known, and there was no difference in the number of steps taken or the time spent per day in moderate-to-vigorous intensity exercise based on BMI category. Most of the girls did not meet the daily physical activity guidelines of 5969–6773 steps/day and 18.5–22.5 min/day of moderate-to-vigorous activity [57]. Further studies using a large cohort need to be carried out to ascertain the differences.

Our findings demonstrated that the majority (90%) of the adolescents in this study spent >4 h/day engaged in sedentary activities on weekdays and 95% did so on weekend days. The participants spent >2 h/day looking at screens on both weekdays and weekend days. Consistent with our findings, Al-Hazzaa et al. [58] conducted a study among 2908 Saudi adolescents aged 14–19 years in the cities of Riyadh, Jeddah and Al-Khobar in Saudi Arabia and found that >91.2% of females and 84% of males used the computer and watched TV >2 h/day. Alharbi’s study on children aged 10–15 years used daily hours spent on electronic devices and TV (≤2 or >2 h) and found that the children who spent ≤2 h/day using these devices were significantly more active than those who spent ≥2 h/day watching TV and using electronic devices [52]. The results of our study correspond with those of Hashem et al. [59] who showed that the mean sedentary time was 568.2 ± 111.6 min/day among 435 Kuwaiti female adolescents on weekdays and weekend days. Only 3.4% of the participants met the physical activity guidelines, and 21% of them met the maximum requirements for sedentary time. Most of the adolescents (79%) exceeded the recommendations for screen time at ≥2 h/day. The boys (74.6%) and girls (83.3%) exceeded the screen time guidelines by 41.4% using only the TV viewing measurement tool [59], which was a key difference compared to our study.

Contrary to our findings, Gaddad et al. [60] revealed that approximately 77% of the Indian participants in this study had low PAQ-A scores, and 83.9% of the girls were relatively more inactive than the boys. Among the girls, physical activity was positively dependent on the ASAQ sedentary and eating attitudes test scores, while the ASAQ score was the only variable related to the PAQ-A score among the males. Furthermore, the female adolescents spent 9.7 h/day engaged in sedentary activities, mostly using small-screen devices. As the number of participants doubled, the percentage of inactive participants was also likely to increase. Moreover, 85.2% of Brazilian adolescent girls had high sedentary behavior levels, and among them, 58.7% used their cell phones for >120 min [48]. However, their sedentary behavior was only assessed over one weekday, and only on-screen time was measured. In contrast, our study evaluated this based on weekdays and weekend days with five categories of sedentary activities. Furthermore, a study by Fletcher et al. [61] among 939 Australian adolescents aged 16–17 years indicated that 69% of these adolescents spent ≥2 h/day using electronic devices, over one third spent ≥2 h/day using computers for recreation, and about 25% watched TV for ≥2 h/day and played e-games. Interestingly, Kerkadi et al. [62] reported that the females in their study spent more time engaged in sedentary activities compared to the males (53.4% vs. 46.4%, respectively), and the mean BMI and WC based on the total screen time (hours/day) among the normal weight participants were 4.82 ± 3.13 and 4.72 ± 3.09, respectively. We further found that the anthropometric measurements used in our study were positively associated with age and negatively associated with sedentary behaviors at weekends. Age and sedentary behaviors were found to significantly (*p* < 0.05) predict the body weight indicators of the study participants (F (2, 396) = 4.346, *p* < 0.014]. A significantly moderate relationship (R = 0.017) was observed between these two predictors (age and sedentary behaviors), Whereas WHtR was positively associated with age. The results were as expected, since adolescence is marked by a rapid pace of growth. In comparison with our current findings, a study conducted among 2906 Saudi adolescents aged 14–19 years found that there was a significant association between overweight/obesity or abdominal obesity and vigorous physical activity levels (adjusted OR for high levels of physical activity = 0.69, 95% CI 0.41–0.92 for BMI and 0.63, 95% CI 0.45–0.89 for waist-to-height ratio) [63]. The multiple linear regression models in this study indicated that self-assessment of physical activity (PAQ-C) did not significantly predict anthropometric measurements (BMI and WHtR; *p* > 0.05). The significant association between objectively measured physical activity (i.e., through an accelerometer) and body weight status was shown in several studies. A study among the Latin American population showed a weak and negative association between BMI and waist circumference, and daily step counts assessed by an accelerometer (*p* < 0.05) [64]. Another study among the Malaysian population showed that physical activity level determined by an accelerometer was significantly associated with the indices of obesity, such as BMI, WC, and body fat percentage [65].

Sociodemographic characteristics, such as participants’ mothers’ and fathers’ education levels and monthly income, were not significantly associated with anthropometric measurements (*p* > 0.05). The descriptive data showed that about half of the participants had educated parents (with tertiary education). The homogeneity of the sample might have contributed to these findings. Nevertheless, the proportion of parents with tertiary education shown in the study was in line with the previous study in Saudi Arabia [66]. The relationship between obesity and sociodemographic characteristics is complex and dynamic. Variables, such as gender, age group, and country under study, affect the nature of the relationship between sociodemographic characteristics and obesity indicators [67].

Many earlier studies have focused on physical and sitting activities but did not cover all the domains of sedentary behavior. The current study is the first to focus on measuring the relationships between physical activity and multiple domains of sedentary behavior and BMI and WC among young female adolescents aged 13–14 years from Arar. We applied a multistage random sampling technique using a probability sampling method during participant selection to eliminate sampling bias. Moreover, the data in this study may provide researchers, healthcare professionals, and nutritionists with information about physical activity and sedentary behavior levels, BMI and WC among Saudi females aged 13–14 years. Government agencies and policymakers could use the findings of this study to develop intervention strategies and public policies related to physical activity and sedentary behaviors among adolescents in Saudi Arabia.

A limitation of this study is that it was only conducted in some schools in Arar and did not include more cities in Saudi Arabia. However, it was not possible to increase the number of schools and cities due to time constraints and the small number of investigators involved in the project. Moreover, Saudi schools for females are separate from those for males, thus male students were not included in the current study. Adding dietary intake as a measurement would give the study more strength and produce more data with regard to the anthropometric measurements. Even though the questionnaire used in the study was validated, the results of the study should be interpreted cautiously. Some limitations need to be considered, and the quality of data collected depended on the ability of the subject to understand the questionnaire, as well as recall and report the data accurately. Lastly, biological maturation variables were not measured, and this could be considered one of the limitations of this study. We recommend this variable to be included in future research among similar group of population.

## 5. Conclusions

This study showed that most of the adolescents aged 13–14 years in Arar were of normal weight; however, the majority of the female adolescents were not meeting physical activity recommendations of at least 420 min of MVPA weekly. The majority of the study participants spent most of their free time engaged in sedentary activities on weekdays and weekend days. Age and sedentary behavior significantly predicted anthropometric indicators among the study participants. Further research is required to ascertain the relationship between physical activity, sedentary behavior, and anthropometric measurements by using objective methods of assessment (i.e., accelerometer) to overcome the limitations of the self-reported technique. In summary, the majority of Saudi female adolescents require more interventions to increase their knowledge, improve their attitudes and change their behaviors toward physical activity and reduce their sedentary behaviors, as this could help them avoid undertaking insufficient levels of physical activity and maintain a normal weight. Accordingly, more educational programs are recommended to prevent female adolescents from becoming overweight or obese as this can lead to NCDs.

## Figures and Tables

**Table 1 ijerph-18-08461-t001:** Sociodemographic and personal characteristics of the participants.

Variable (s)	Number/Mean ± SD	Percent (%)
Age group (years)	13.34 ± 0.49	
School location		
Central region	111	27.8
East region	74	18.5
North region	110	27.6
West region	104	26.1
Number of siblings		
<5	169	42.4
5–10	212	53.1
>10	18	4.5
Number of household members		
<5	17	4.3
5–10	308	77.2
>10	74	18.5
Mother’s education		
Intermediate school or lower	85	21.3
High school	122	30.6
Undergraduate or higher	192	48.1
Father’s education		
Intermediate school or lower	50	12.5
High school	131	32.8
Undergraduate or higher	218	54.6
Family monthly income (* SAR)		
<5000	57	14.3
5000–14,999	198	49.6
≥15,000	144	36.1

* 1 USD = 3.75 SAR (Saudi Arabia Riyal).

**Table 2 ijerph-18-08461-t002:** Anthropometric characteristics of the respondents.

Variable (s)	(Mean ± SD)	Frequency	Percentage (%)
Height (m)	151.28 ± 9.51		
Weight (kg)	51.06 ± 13.62		
BMI (kg/m^2^)	22.92 ± 14.16		
BMI z-score	0.57 ± 1.39		
WC (cm)	70.7 ± 10.1		
WHtR	0.46 ± 0.06		
BMI status			
Thinness < −2 SD		20	5.0
Normal ≤ −2 SD and ≥ 1 SD		317	79.4
Overweight > +1 SD		52	13.0
Obese > +2 SD		10	2.5
WC			
Normal		249	62.4
Overweight 72.3 cm		52	13.0
Abdominal Obesity 77 cm		98	24.6

WHtR			
WHtR < 0.5		297	74.4
WHtR ≥ 0.5		102	25.6

Abbreviations: BMI—body mass index; WHtR—waist to height ratio; WC—Waist circumference.

**Table 3 ijerph-18-08461-t003:** Distribution of participants by physical activity level.

Variable (s)	Frequency	Percentage (%)
Not meeting physical activity recommendations (<420 min weekly)	370	92.7
Meeting physical activity recommendations (≥420 min per week)	29	7.3

**Table 4 ijerph-18-08461-t004:** Participants’ sedentary behavior during weekdays and weekends.

Sedentary Behaviors	(Mean ± Standard Deviation) (Minutes)	Frequency	Percentage (%)
ASAQ Score Weekdays	357.64 ± 86.29		
ASAQ Score Weekend days	470.51 ± 147.64		
Weekdays			
Screen time (min/day)	141.03 ± 58.69		
Education (min/day)	117.89 ± 36.22		
Travel (min/day)	17.95 ± 8.65		
Cultural (min/day)	37.02 ± 28.73		
Social (min/day)	44.94 ± 18.48		
Low < 4 h		40	10
High ≥ 4 h		359	90
Weekend days			
Screen time (min/day)	201.37 ± 97.37		
Travel (min/day)	23.68 ± 14.84		
Cultural (min/day)	59.95 ± 46.46		
Social (min/day)	68.77 ± 39.89		
Low < 4 h		20	5
High ≥ 4 h		379	95

**Table 5 ijerph-18-08461-t005:** Relationship between sociodemographic characteristics, physical activity, and sedentary behavior with BMI and WHtR, explored using simple and multiple regressions analyses.

Variable	Adjusted (Multiple Regression)—BMI			Adjusted (Multiple Regression)—WHtR		
	B (β) (95% CI)	*p* Value	VIF	B (β) (95% CI)	*p* Value	VIF
Sociodemographic characteristics						
Age (years)	3.523(0.661, 6.386)	0.016	1.011	0.198 (0.127–3.709)	<0.001	1
Number of siblings						
Number of household members						
Mother’s education						
Father’s education						
Family’s monthly income						
Physical activity						
Total mean score of physical activity						
Sedentary behavior						
Total sedentary behavior weekdays						
Total sedentary behavior weekends	−0.009 (−0.019, 0.000)	0.055	1.011			
F value	4.346			16.191		
*p*-value	0.014			0		
Adj R^2^	0.017			0.037		

Abbreviations: BMI—body mass index; WHtR—waist to height ratio; β—regression coefficient; CI—confidence interval; VIF—variance inflation factor.

## Data Availability

The data presented in this study are available on request from the corresponding author.

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
