# Peer review of "Relationship between Physical Activity, Sedentary Behavior, and Anthropometric Measurements among Saudi Female Adolescents: A Cross-Sectional Study"

_ijerph, 2021, doi:10.3390/ijerph18168461_

Round 1

Reviewer 1 Report

This paper descibes in teenage girls the relationship between physical activity and physical inactivity and overweightness.

The paper is a crosssectional study and cannot use , as in abstract, results and conclusion the suggestion of a causal relationship ( f.i. in conclusion physical activity and inactivity inluences body weight indicators.

In tables  the results are given in three or four decimals behind the comma and this suggests a non existing appearenses.

I suggest in table 5 not to give the unadjusted reults ( say it in words) but only the adjusted results)

Reviewer 2 Report

Congratulations to the authors for their work. The quality of the article has improved considerably. The fact that we have analyzed compliance with the new MVPA recommendations (through the PAQ-A) makes the article suitable for citation and comparison with future studies. I leave some minor modifications that could improve it even more:

  1. Title: To me, 'body weight indicator' is wrong. 'Anthropometric measurements' could be an option. Another option could be 'obesity-related parameters'. Please, modify in all the manuscript. 
  2. I recommend that authors replace waist circumference with waist-to-height (>0.5) to determine abdominal obesity. This indicator is more useful and accurate.  
  3. All tables should include additional information about the expression of the data, acronyms, or whether the data are adjusted. Tables should be understandable on their own. 
  4. Table 5 refers to BMI, they could make another table (or add in table 5) the results for waist circumference (or even better for WHtR).
  5. The article must be reviewed by a native speaker.

Kind regards,

Reviewer 3 Report

The topic addressed by the authors in this cross-sectional study is very interesting. The problem of overweight and obesity in young people is now global. Therefore, the authors with their study highlighted the need to implement programs to improve well-being and lifestyle in order to prevent the risk of contracting non-communicable diseases.

Abstract

The abstract does not follow the editorial standard indicated for IJERPH: “The abstract should follow the style of structured abstracts, but without headings”; therefore, remove the words: Background, Objective, Results and Conclusions.

Keywords

To optimize the search of the manuscript on the search engines, insert different keywords from those present in the title.

Remove “physical activity”, “sedentary behaviour”, “body weight indicator”, "Saudi female adolescents" and, if necessary, add other relevant key words. It is important to add words other than those present in the title.

I have no other comments to make as the study is methodologically sound.

Reviewer 4 Report

Thank you very much for the opportunity to review this paper. Many thanks to the authors for the work presented and the time they have taken for it. 

This is a paper that analyzes the relationship between physical activity, sedentary lifestyle and weight in female adolescents in Saudi Arabia. 

The abstract covers the most important points, although they should indicate the statistical analyses performed. At least the most important ones. 

The introduction begins by highlighting the levels of obesity and providing global data. Similarly, the following paragraphs deal with "insufficient levels of physical activity" and "levels of sedentary lifestyles". The introduction should be improved with a more in-depth analysis of the variables instead of just providing data. Rather than an introduction, it seems more like a discussion section in which the results of different research studies are contrasted.

Regarding the methodology, the description of the sample is very insufficient. They should provide data such as, for example, the hours of physical education they do at school, since this is something that all of them do. They could also indicate the socioeconomic level of the families and whether they live in rural areas or in cities, since these are variables that affect the results they intend to show. 

The collection of data in the survey does not guarantee that the answers are true since an interview with the researchers may cause the respondents to indicate the "desired answer". Furthermore, there is no indication that there is any other technique for checking the veracity of the data or cross-checking with other sources. 

The statistical analysis performed was a descriptive analysis and a multiple linear regression model. I consider these to be insufficient for the quality of the studies that should be presented in a journal of this level of impact. 
Regarding the results, Table 1 shows the sociodemographic characteristics. These data should be included in the section on the sample.

The conclusion should respond to the objective indicated in the paper and should not present any data. The numbers should appear in the results section and not in the conclusions where they should be affirmed or denied in an important way according to the results of the study. 

Round 2

Reviewer 1 Report

Authors have changed their manuscript according my suggestions and revised manuscript can be accepted for publication

Reviewer 4 Report

The authors have done a good job of adapting the paper to the requirements of the reviewers. 
In this sense, I consider that the work still needs to make minor corrections. That is why in the introduction they should somehow connect the variables used in their study with the concept of quality of life. That is why the following reference may help you: 
Arcila-Arango, J. C.; Castro-Sánchez, M.; Zagalaz-Sánchez, M. L.; Valdivia-Moral, P. (2021). Analysis of quality of life in Colombian university students. Journal of Sport and Health Research. 13(2):295-304.
https://recyt.fecyt.es/index.php/JSHR/article/view/89606

On the other hand, it is important in the discussion to compare with studies from other countries. That is why it is recommended to review this good study conducted throughout Latin America: 
Ferrari, G.; Marques, A.; Barreira, T.V.; Kovalskys, I.; Gómez, G.; Rigotti, A.; Cortés, L.Y.; García, M.C.Y.; Pareja, R.G.; Herrera-Cuenca, M.; Guajardo, V. Leme, A.C.B.; Guzmán Habinger, J.; Valdivia-Moral, P.; Suárez-Reyes, M.; Ihle, A.; Gouveia, E.R.; Fisberg, M.; on behalf of the ELANS Study Group. Accelerometer-Measured Daily Step Counts and Adiposity Indicators among Latin American Adults: A Multi-Country Study. Int. J. Environ. Res. Public Health 2021, 18, 4641. https://doi.org/10.3390/ijerph18094641 

Finally, with regard to the conclusions, I believe that the authors should emphasize even more how the data of their work respond to the objectives of the study. 

Author Response

This manuscript is a resubmission of an earlier submission. The following is a list of the peer review reports and author responses from that submission.

Round 1

Reviewer 1 Report

This manuscript analyses the relation between anthropometric variables of body composition (BMI and WC) in13-14 years old girls with  physical (in)activity measured by questionnaires. Also income, family composition and education of the parents are included.

However the most important variable as biological maturation is not measured. Calendar age is in this age group not important.

The stage of maturation has to be included as co-variable in the multiple regression.

Also not clear is how BMI and WC are combined in the statistical analysis of table 5.

Author Response

Dear Professor,

Thank you for giving me the opportunity to submit a revised draft of my manuscript entitled “Relationship Between Physical Activity, Sedentary Behaviour and Body Weight Indicators among Saudi Female Adolescents : A Cross-sectional Study” . We are grateful to the reviewers for their insightful comments on the manuscript. We have improved the manuscript and highlighted the changes within the manuscript. We also listed a point-by-point response to the reviewers’ comments and concerns.

We look forward to your favourable response and thank you in advance for your time and consideration.

Sincerely,

Kind regards

Assoc. Prof Dr Hazizi Abu Saad

Reviewer 2 Report

Thank you so much for inviting me to review this manuscript. This study tried to determine the association between physical activity 14 level, sedentary behaviour and body weight indicators in Saudi female adolescents age 13 and 14 years from Arar. I have to reject this article. While it has some interesting aspects, it needs a lot of changes (especially methodological) in order to be considered for publication. Here are some comments: 

  • The authors have not performed the statistical analysis segmented by sex. It has been shown in a number of scientific articles that girls are less active than boys (which could seriously influence both the results obtained and the conclusions drawn).

  • The authors have reported the names of the schools analysed in Table 1. In my opinion, this should be removed in order to guarantee the anonymity of the participants. It is more than sufficient to indicate the region of the country where the survey was conducted.

  • To me, the term 'physical inactivity' is not correct. It is more appropriate to use 'insufficient levels' of physical activity. Not meeting with the recommendations does not mean that there is no movement.

  • Alharbi (2019) used cut-off points that have not been validated for the study population. The cut-off points of Voss et al. have been validated in a different geographical location, population and culture. It is not valid to apply these cut-off points. I recommend the authors to read this manuscript: https://doi.org/10.3390/children7120263. It is possible to obtain MVPA minutes and the meeting of the PA recommendations through PAQ-C score (which is more appropriate than applying these cut-off points).

  • Finally, I am not  an expert with English. However, the article has serious grammatical errors with English that I have noticed. The article should be checked by a native translator.

Best wishes,

Author Response

(The authors gave the same response as above.)

Reviewer 3 Report

  • I suggest including the type of study in the title (: a cross-sectional study). I usually appreciate this information when selecting publications for consultation and citation.
  • Line 67-83- I suggest including a reference to what is considered (parameters , Mets or hours/day) sedentary behavior.
    There are attitudes that can define / contribute to sedentary attitudes (watching TV, computer, video games, tablets). Talk a little more about it with references. It is even important to contextualize this with the Saudi reality.
  • 2.2 - Line 96- I suggest to include that "The entire development of the study followed the ethical recommendations determined by the Declaration of Helsinki."
  • Define SD in the legend of table 2.
  • Describe PAQ-C Score in the legend of table 3.
  • Describe ASAQ in the legend of table 4.
  • I did not identify the associations between body weight indictor (WC and BMI) in table 5.
  • Line 331-334 - I understand that this must be the strength of your work. However, it is necessary to say what this information will be used for. thinking about intervention strategies and public policies, how can these results positively impact health actions, reduce the BMI of adolescents and promote a better quality of life? I suggest making a separate paragraph, with the subtitle - The strength. 
  • Line 334-340 - I suggest a separate paragraph to state the limitation of this study (and not part). You only learn one limitation, and that doesn't check part of the limitations. It's just one. But I believe that it should be separated from the previous text, in a prominent paragraph.

Author Response

(The authors gave the same response as above.)

Reviewer 4 Report

Congratulations to the authors for a good job. I only have brief recommendations.

Introduction

The authors should reflect some hypothesis of the study and describe the objectives in greater detail.

Study participants

Why was the sample only made up of 13 and 14 year old adolescents? It is a very short age range.

Conclusions

It is recommended to reflect the implications for practice of this study.

Author Response

(The authors gave the same response as above.)

Round 2

Reviewer 2 Report

Thank you for inviting me to review this manuscript again. The authors have modified some aspects that I indicated. However, some issues have not been resolved.

Point number 2. I am sorry. I have revised my last comment on this indication. I will try to explain this question. Was the research conducted on men and women? If so, why did it not include men in the study? And if, on the other hand, it was only conducted on women, what is the justification for this choice? I hope this is now clear and I am sorry for the confusion. 

Point number 5. On the other hand, what worries me most are the cut-off points applied. These cut-off points used by the authors are not validated and are totally arbitrary. Alharbi (2019) indicates 'The scores were also categorized as low (≤2.3), moderate (2.4-3.7), and high (≥3.8) levels of physical activity'. Why did he use these cut-off points? Where are these cut-off points validated? Just because these criteria have been used in a similar sample does not mean that they offer validity for application. On the contrary, the study I mentioned, determines the weekly minutes of MVPA (according to the new physical activity guidelines of the World Health Organisation (please see Bull et al., 2020) through the PAQ-C and PAQ-A scores. This is not a very complicated task to perform and I believe it could considerably improve the quality of the mentioned article (even if it is disputed with other studies that applied PAQ-C/PAQ-A but did not use this methodology. In my opinion, it does not make sense to apply unvalidated cut-off points when there is another methodology available that is more appropriate and that allows to determine the meeting of the  new physical activity guidelines published by the World Health Organization (2020). Furthermore, the authors refer to Voss et al. In that study: 'The sample was drawn from the East of England Healthy Hearts'. Do the authors think that the Saudi female adolescents are similar (in terms of physical activity level) to those in that study? These criteria cannot be applied to the study population.

From here, two options: Either organise the sample by tertiles (and indicate the limitation of not being able to apply cut-off points) (less recommended option). Or apply the study methodology proposed in Saint-Maurice et al. (2014) (Calibration of self-report tools for physical activity research: the Physical Activity Questionnaire (PAQ)) as previously indicated (highly recommended option).

Without either of these modifications I am unable to accept the article for publication.

Best wishes,